# The Role of Dioxygen in Microbial Bio-Oxygenation: Challenging Biochemistry, Illustrated by a Short History of a Long Misunderstood Enzyme

**DOI:** 10.3390/microorganisms12020389

**Published:** 2024-02-15

**Authors:** Andrew Willetts

**Affiliations:** 14 Sv Ivan, 21400 Sutivan, Croatia; andrewj.willetts@btconnect.com; 2Curnow Consultancies, Helston TR13 9PQ, UK

**Keywords:** dioxygen, molecular oxygen, Baeyer–Villiger mono-oxygenase, 2,5-diketocamphane mono-oxygenase, flavin-dependent two-component monooxygenase, flavin reductase, putidaredoxin reductase, *Pseudomonas putida* ATCC 17453

## Abstract

A Special Issue of *Microorganisms* devoted to ‘Microbial Biocatalysis and Biodegradation’ would be incomplete without some form of acknowledgement of the many important roles that dioxygen-dependent enzymes (principally mono- and dioxygenases) play in relevant aspects of bio-oxygenation. This is reflected by the multiple strategic roles that dioxygen -dependent microbial enzymes play both in generating valuable synthons for chemoenzymatic synthesis and in facilitating reactions that help to drive the global geochemical carbon cycle. A useful insight into this can be gained by reviewing the evolution of the current status of 2,5-diketocamphane 1,2-monooxygenase (EC 1.14.14.108) from (+)-camphor-grown *Pseudomonas putida* ATCC 17453, the key enzyme that promotes the initial ring cleavage of this natural bicyclic terpene. Over the last sixty years, the perceived nature of this monooxygenase has transmogrified significantly. Commencing in the 1960s, extensive initial studies consistently reported that the enzyme was a monomeric true flavoprotein dependent on both FMNH_2_ and nonheme iron as bound cofactors. However, over the last decade, all those criteria have changed absolutely, and the enzyme is currently acknowledged to be a metal ion-independent homodimeric flavin-dependent two-component mono-oxygenase deploying FMNH_2_ as a cosubstrate. That transition is a paradigm of the ever evolving nature of scientific knowledge.

## 1. Introduction

Dioxygen (O_2_, more commonly referred to as molecular oxygen) is the only element in the Earth’s environment that is paramagnetic in its ground-state [1]. This energetically favoured state is determined by the molecule preferentially existing as a triplet diradical (^3^Σ), with its two least strongly held electrons, the ones most responsible for its chemistry, occupying two separate spin-paired anti-bonding π* orbitals [2]. This in turn dictates that a concerted reaction between O_2_ and carbon in organic compounds is spin-forbidden [3]. However, evolution has generated a significant number of enzymes able to oxygenate organic substrates by the prior activation of molecular oxygen to a 1-electron reduced form (O_2_^−^), either by coopting an organic radical (e.g., a flavin nucleotide), or deploying a transition element (characterised by partially filled outer-bonding orbitals), which may (e.g., heme-coordinated iron) or may not (e.g., nonheme iron) be coordinated into an organic cofactor [4,5].

## 2. The Early Years

While the first indirect evidence of the involvement of molecular oxygen in an enzyme-catalysed reaction emerged in the mid-1930s from studies of an FMN-dependent NADPH oxidoreductase (Old Yellow Enzyme) isolated from the bottom of fermenting yeast *Saccharomyces carlesbergensis* [6], it was over two decades later before direct confirmation of bio-oxygenation reactions catalysed by microbial enzymes was advanced as a result of similar programmes of research conducted by Osamu Hayaishi and Howard Mason. Hayashi used mass spectroscopy to established that an enzyme he confirmed contained a nonheme atom of the transition element iron (Fe^3+^) was able to incorporate both atoms of ^18^O_2_ in biotransforming catechol to cis,cis-muconate (Figure 1, [7]). He initially termed the enzyme, sourced from an environmental microbial isolate, ‘pyrocatechase’. Concurrently, Mason used the same technique to confirm that an enzyme he called ‘mushroom phenolase’, which was devoid of any form of iron but did contain a bound atom of the transition element copper (Cu^2+^), was able to biotransform 3,4-dimethylphenol to 4,5-dimethylcatechol by incorporating a single atom of ^18^O_2_ [8]. In an attempt to distinguish between the two enzyme types, Mason subsequently suggested that the terms ‘mixed-function oxidase’ and ‘oxygen transferase’ should be used to discriminate between the respective modes of action of ‘mushroom phenolase’ and ‘pyrocatechase’ [9]. However, this proposal was not received favourably by Hayaishi, who countered by advancing the corresponding names ‘monooxygenase’ and ‘dioxygenase’ to better reflect the differing outcomes of these two types of O_2_-dependent enzymes [10]. Significant in that respect, Hayaishi’s functionally more relevant terms have stood the test of time, and are still widely used decades later to describe many of the other disparate examples that have been subsequently recognised [11]. Semantic disagreements apart, Mason and Hayaishi’s pioneering studies promoted a substantial body of ongoing research to better characterise the biochemistry deployed by the two enzyme types in facilitating their respective interactions with molecular oxygen. However, a fully comprehensive understanding for each enzyme type has proved illusive, with a number of aspects still currently remaining works in progress. This is well illustrated by the reviewing the evolution of the perceived mode of action of 2,5-diketocamphane 1,2-monooxygenase (2,5-DKCMO). This monooxygenase is one of a suite of inducible enzymes coded for exclusively by corresponding genes located on the CAM plasmid of *Pseudomonas putida* ATCC 17453 [12,13,14,15] that collectively serve to biodegrade camphor to central intermediary metabolites (Figure 2, [16]). While 2,5-DKCMO has been studied for over 60 years, some aspects of the relationship between the enzyme and dioxygen remain incompletely characterised in 2024.

## 3. The Rise and Subsequent Fall of a Metal Ion-Dependent Monooxygenase

2,5-DKCMO is notable for being the O_2_-dependent activity that serves a key role in triggering the initial ring-opening step in the pathway of bicyclic (+)-camphor catabolism. Additionally, it has one indisputable claim to fame. It was the first enzyme reported to function as a biological Baeyer–Villiger monooxygenase (BVMO, [17,18,19]). BVMOs are so named because they bio-oxygenate ketones into corresponding lactones/esters, a transformation directly equivalent to the peracid-catalysed chemical reaction first reported by Adolf Baeyer and Victor Villiger in 1899 [20]. Although the lactonization by various bacteria and fungi of the D-ring of 4-androstene-3,17-dione to testololactone had been recognised by 1953 [21,22], the nature of the enzyme(s) responsible remained uncharacterised for several more years. Then, subsequently, definitive proof of the role of molecular oxygen in another microbial lactonization, the initial ring-opening step of camphor degradation by *P. putida* ATCC 17453, was obtained in late 1961 by Irwin Gunsalus at the University of Illinois [23]. He confirmed that a cell-free extract prepared from the (+)-camphor-grown bacterium biotransformed 2,5-diketocamphane (2,5-DKC) into the corresponding 2-oxa-lactone (Figure 2, [16]) ‘only when the extract was supplemented with NADH in the presence of oxygen’. Gunsalus initially used the term ‘ketolactonase, an enzyme for cyclic lactonization’ to describe the detected O_2_-dependent monooxygenase activity which is currently assigned as EC 1.14.14.108.

Triggered by this initial observed outcome, Gunsalus then focussed his attention on establishing in more detail the mode of action of the ketolactonase that resulted in it being able to catalyse the spin-forbidden O_2_-dependent biotransformation of the DKC enantiomer. At the time, the precedents set by other O_2_-dependent enzymes had identified a number of alternative cofactor-dependent strategies for enabling inert molecular oxygen to participate in one electron-state biochemistry. These included the seminal recognised deployment of nonheme iron (catechol 1,2-dioxygenase, [7]), or copper (3,4-dimethylphenol monooxygenase, [8]), as well as other less directly confirmed options that were dependent on either heme-coordinated iron (tryptophan 2,3-dioxygenase, [24]), or FAD (D-amino acid oxidase, [25]). The then current protocols for discriminating between these alternative possibilities were based on either characteristic absorbance spectra (both flavin nucleotide- and heme iron-dependent enzymes), or the effect of metal chelating agents (iron- and copper-dependent enzymes, [26]). Also used were some electron acceptors such as methylene blue and dichlorophenolindophenol, which had been used to indirectly signal the involvement of a flavin nucleotide in the action of Old Yellow Enzyme [6]. Compared to more modern analytical methodologies for characterising O_2_-dependent enzymes such as Mὄssbauer and EPR spectroscopy [27], such techniques were rudimentary and non-specific, leaving outcomes open to alternative possible explanations.

Gunsalus initially established [28,29,30,31,32] that the lactonizing activity was dependent on the combined activities of two enzymes (E_1_ and E_2_) which upon purification were both confirmed to bind FMN. E_1_ was monomeric, (estimated MW 50,000 kDa), bound NADH but not NADPH, had an absorption spectrum that did not exhibit a characteristic heme-generated Soret band in response to CO, was not inhibited by divalent metal ion chelating agents (bipyridyl = Fe^2+^; NaN_3_ = Cu^2+^), and was able to donate electrons to decolourize methylene blue. Conversely, E_2_ was also reported to be monomeric (estimated MW 80,000 kDa), did not exhibit a Soret band or the ability to reduce methylene blue, but was strongly inhibited by bipyridyl but not NaN_3_. Gunsalus interpreted this initial data to conclude that both E_1_ and E_2_ were flavoproteins, both physically and functionally linked as a multienzyme complex in which the flavin nucleotide served as a molecular bridge or conduit to channel reducing power between the two activities. E_1_ was termed NADH oxidase, to reflect Mason’s terminology [9], and functionally served to reduce FMN to FMNH_2_ independent of any involvement of either Cu^2+^, Fe^2+^, or heme iron. Conversely, its partner enzyme E_2_ was a monooxygenase independent of Cu^2+^ and heme iron, but dependent on Fe^2+^ (nonheme iron) to promote the FMNH_2_ + O_2_-dependent lactonization of DKC. These outcomes were summarised as a simple outline correlation (Figure 3A). Subsequently, as a result of a number of additional publications [33,34,35,36,37], the roles of E_1_ and E_2_ were further refined, including more detail of the predicted key valency changes undergone by the nonheme iron content of E_2_ (Figure 3B). While the last of that sequence of publications in September 1966 still maintained a pivotal role for the interchange of Fe^2+^/Fe^3+^ in the functioning of E_2_, that all changed in late 1969 when Gunsalus published a short paper most significant for reporting that, contrary to his previous proposals, the monooxygenase E_2_ was bipyridyl-insensitive, and consequently concluded to be metal-free [38]. It may be no coincidence that the late 1960s corresponds with the commencement of Gunsalus’ collaboration with Helmut Beinert and William Orme-Johnson in the Physics Department at Illinois. These researchers were experts in deploying highly sensitive EPR and Mössbauer spectroscopic techniques and interpreting the resultant hyperfine resonance signals generated. Significantly, these techniques could be used to confirm the presence in the proteins of embedded atoms with a magnetic nucleus [39]. By modifying the salt composition of the camphor-based culture medium of *P. putida* ATCC 17453 to replace ^56^Fe (nuclear spin zero) with ^57^Fe (magnetic nucleus), this would have enabled Gunsalus to establish directly whether the induced 2,5-DKCMO contained any iron of any description. One can only speculate that Gunsalus searched for and failed to find such a corresponding signal, an outcome which would have proved transformational, thereby prompting his 1969 publication [38]. Gunsalus offered no alternative explanation for the implied iron-free activation of dioxygen by E_2_, a deficiency which seriously undermined his previous 1961–1966 mantra. It is also notable that no subsequent reported research on 2,5-DKCMO was undertaken by Gunsalus, who instead swopped his attention to investigating the involvement of iron in the bioactivation of O_2_ for cytochrome P450 monooxygenase [40], the key multicomponent bio-oxygenating enzyme that initiates the camphor biodegradation pathway of *P. putida* ATCC 17453 (Figure 2).

## 4. The Fallow Years

After a hiatus of over a decade, interest in 2,5-DKCMO was rekindled at the University of Aberystwyth by Peter Trudgill, one of Gunsalus’ former collaborators at Illinois. Significantly, in the intervening years several relevant things had changed, including better ways of purifying and characterising enzymes (e.g., gel-filtration and ligand-affinity chromatography, plus SDS-PAGE and Phast gel electrophoresis). Also highly relevant was that studies driven by Vincent Massey [41], David Ballou [42,43], and Thomas Bruice [44,45] had led to a better understanding of the interactions between reduced nicotinamide nucleotides and flavin nucleotides that serve to activate molecular oxygen by forming key C4a-peroxy- and C4a-hydroperoxyflavin moieties which could then drive bio-oxygenations reactions (Figure 4, [46]). Despite these various advances, most of Trudgill’s limited output over the period of 1982–1993 [47,48,49,50,51,52,53] had more than a hint of déjà vu, being essentially a re-examination of structural aspects of 2,5-DKCMO (E_2_) and its associated NADH oxidase activity (E_1_). Notable was that while highly purified samples of both E_1_ and E_2_ were confirmed to bind FMN and be devoid of nonheme iron, no proposals were advanced to explain how the necessary activation of molecular oxygen was achieved to ensure the bio-oxygenating role of E_2_. Trudgill merely reiterated Gunsalus’ mantra that both E_1_ and E_2_ were true flavoproteins, and as such were dependent on bound flavin mononucleotide serving as a cofactor for activity. Again, while titration with purified samples of E_1_ and E_2_ confirmed Gunsalus’ proposal that they formed a multienzyme complex in an equimolar ratio, no suggested explanation of the functional interaction of the two enzymes was advanced. Two different methods established that the MWs of E_1_ and E_2_ were 36,000 kDa and 78,000 kDa, respectively, both values lower than those reported previously by Gunsalus. Also, for the first time, SDS-PAGE analysis confirmed that E_2_ was a homodimer of two apparently identical subunits, although somewhat surprisingly, the same technique was not applied to E_1_ (vide infra).

## 5. New Insights: The Relevance of the Correlation with Other Classes of Monooxygenase

At around the same time, an interdisciplinary team at the University of Exeter led by Stanley Roberts began using the NADH-dependent 2,5-DKCMO as a biocatalyst to generate key synthons for valuable chemoenzymatic syntheses [54,55,56,57,58,59]. While NADPH-dependent cyclohexanone monooxygenase (CHMO) from the *Acinetobacter* sp. strain NCIB 9871 had previously been developed successfully as such a lactone-generating enzyme [60], the significantly cheaper cost of NADH was the principal driver for this Exeter initiative. A complementary screening programme run by Raffaella Villa was also undertaken to include other known potentially useful NADH-dependent bio-oxygenating enzymes [61]. One important outcome from this approach was to establish a number of functional and homology similarities between the ketolactonase and a number of previously well-studied bacterial luciferases. Strongly supporting this recognised relationship was the novel reported ability of the luciferases, whose natural evolved substrate is the aliphatic aldehyde dodecanal [62], to lactonize the same abiotic alicyclic ketones as 2,5-DKCMO. As a result, for the first time, these enzyme types were grouped together as Type II BVMOs [63] to distinguish them from the corresponding NADPH-dependent Type I flavoprotein enzymes such as CHMO. While the Exeter studies confirmed that both Type II BVMO enzyme types shared a common dependence on NADH and FMNH_2_, the significance for 2,5-DKCMO of the previously established deployment by the luciferases of the reduced flavin nucleotide as a cosubstrate rather than an enzyme-bound cofactor [62] was not fully appreciated at the time. Indeed, it would be several years later before this initial delineation of Type II BVMOs as a related group of lactone-generating enzymes served as a key foundation for the emergence of a sounder understanding of the biochemistry of the interaction between 2,5-DKCMO and dioxygen.

The initial catalyst to better comprehending that biochemistry was a review by Holly Ellis [64] that served to relate the FMN-dependence of known bacterial luciferases to a small number of other functionally equivalent recently recognised bacterial monooxygenases. Ellis introduced the term two-component flavin-dependent monooxygenases (fd-TCMOs) to define the new grouping. Most significantly, fd-TCMOs are not true flavoproteins. Both structurally and functionally, they are clearly unrelated to the canonical flavoprotein monooxygenases such as CHMO from the *Acinetobacter* sp. strain NCIB 9871 that deploy a bound flavin nucleotide cofactor [60]. Rather, fd-TCMOs operate as two clearly distinct half-reactions, with a flavin reductase (FR) serving to reduce FMN as a substrate to FMNH_2_ which is then transferred to serve as a cosubstrate for a monooxygenase which catalyses the bio-oxygenating activity of the second half-reaction (Figure 5). Some of these FRs are flavoproteins that operate a ping-pong mechanism to reduce a second external FMN substrate, whereas others are not, being nonflavoprotein reductases that reduce the FMN substrate by a sequential mechanism (Figure 6). The mechanism of transfer of the FMNH_2_ cosubstrate to promote the oxidative half-reaction also discriminates between those fd-TCMOs that depend on close protein–protein interaction between the two participating the enzymes catalysing the two half-reactions, and those that rely on diffusion between more distal separate participating enzymes. Also highly significant is that whereas for some fd-TCMOs a relevant FR is expressed in close proximity on the same operon as the monooxygenase partner, most probably for logistical reasons [65,66,67,68], in other cases [69,70,71,72] the monooxygenase deploys an FMNH_2_ cosubstrate generated by one or more distally located ‘housekeeping FRs’ [73,74,75,76,77], so called because they characteristically share multiple additional redox roles within the same bacterium, including protection against osmotic [78] and oxidative stress [79,80].

It was against this background that Uwe Bornscheuer at the University of Greifswald initiated a research programme in 2011 to capitalise on the BVMO-based interdisciplinary research undertaken at Exeter in the 1990s. Fundamental to the initiative was the cloning and recombinant expression in *E. coli* of a sequenced 2,5-DKCMO gene (4485 bp; NCBI GenBank AY450285) sourced from *P. putida* ATCC 17453 in order to generate significant amounts of the monooxygenase for subsequent development into a coupled-enzyme biocatalytic tool [81,82]. Because the gene coding for the 36 kDa NADH oxidase of *P. putida* ATCC 17453 claimed by both Gunsalus [28,29,30,31,32,33,34,35,36,37,38] and Trudgill [47,48,49,50,51,52,53] to be the true corresponding redox partner for 2,5-DKCMO was not included in the cloned construct, it was anticipated that any expressed monooxygenase would have little or no corresponding activity without the addition of an exogenous source of FMNH_2_. However, extracts of the transformed bacterium were found to generate significant levels of corresponding bio-oxygenated products when challenged with a range of putative ketone substrates. Aware of Ellis’s 2010 review [64], Bornscheuer interpreted this serendipitous outcome as the first albeit indirect evidence that 2,5-DKCMO was not a flavoprotein deploying FMNH_2_ as a bound cofactor as Gunsalus’ prevailing mantra had dictated for 50+ years (vide supra). Rather, like the luciferases and the small number of recently identified bacterial monooxygenases cited by Ellis, the ketolactonase was actually a member of the fd-TCMOs coopting FMNH_2_ as a cosubstrate partner for 2,5-DKC. Although unable to confirm his proposal directly, Bornscheuer speculated that in this particular case the source of the FMNH_2_ that was promoting the bio-oxygenating activity of the overexpressed 2,5-DKCMO was most likely to be one or more of the native ‘housekeeping FRs’ coded for by the chromosomal DNA of *E. coli*. The chromosome of the enteric bacterium codes for a number of such FRs of which the best characterised are Fre [83,84] and HpaC [65,85]. Significantly, it was this important change of mindset by Bornscheuer that first extended the apparent functional similarity between the ketolactonases and bacterial luciferases (Type II BVMOs) to additionally include the bacterial fd-TCMOs. The impact of this proposal was profound, and ultimately proved to be the catalyst that prompted others to initiate the development of the current revised understanding of the biochemistry that defines the relationship between ketolactonases such as 2,5-DKCMO and dioxygen.

## 6. Confirming the Role of 2,5-DKCMO as a Two-Component Monooxygenase

Direct evidence to support the new status of 2,5-DKCMO then followed from a study undertaken by a multinational research group led by Peter Lau that deployed genomic tools to further characterise the biochemistry of some of the key enzymes of the camphor biodegradation pathway of *P. putida* ATCC 17453 [86]. Notably, the group published a seminal 2013 study which specifically achieved two principal goals with respect to 2,5-DKCMO [87]. Firstly, for the first time, the complete sequence of the DNA of the 533 kb double-stranded linear CAM plasmid [12,13,14,15] was established, and homology searching was used to assign functions to every one of its *orf*s. This was an important outcome because it confirmed directly that the plasmid encodes relevant genes for all the characterised enzymes of both the (+)-camphor (2,5-DKCMO-dependent) and (-)-camphor (3,6-DKCMO-dependent) degradation pathways (Figure 2). Further, for the first time it became apparent that 2,5-DKCMO functions as a mixture of two very similar isoenzymic forms (60% overall similarity) coded for by two genes (*camE*_25-1_ and *camE*_25-2_) located on opposite stands of the plasmid. Secondly, the 2013 study identified and characterised Fred, a chromosomal DNA-coded 2 × 18 kDa homodimeric nonflavoprotein reductase that reduced FMN as an external substrate (Figure 6). Both structural homology and mode-of-action studies confirmed that Fred closely resembled a number of other characterised FRs confirmed to support the activity of other known bacterial fd-TCMOs [65,66,67,68,69,70,71,72]. Significantly, cloned and expressed Fred was confirmed to serve as a source of the FMNH_2_ cosubstrate necessary for cloned and overexpressed copies of both 2,5-DKCMO isoenzymes to function effectively with a wide range of alicyclic substrates. For the first time in over 50 years since initially studied by Gunsalus in the 1960s, the collective outcomes of Lau’s 2013 seminal study convincingly demonstrated that CAM plasmid-coded 2,5-DKCMO was a not a flavoprotein dependent on FMNH_2_ serving as a bound cofactor, but rather a fd-TCMO coopting the reduced flavin nucleotide as a cosubstrate generated by a distal chromosome-coded nonflavoprotein reductase. In one final bizarre twist, Lau and others who contributed to that seminal 2013 publication were then subsequently cited as coauthors of a 2015 structural study of 3,6-DKCMO (*camE*_36_, Figure 2) led by Jennifer Littlechild [88]. Not only did the study consistently fail to acknowledge and take into account the confirmed cosubstrate status of FMNH_2_ for the isoenzymic DKCMOs of ATCC 17453 [89], but it also included the extraordinary claim to have ‘now identified a flavin reductase adjacent to the 3,6-DKMO gene on the CAM plasmid’. This unsupported claim was clearly incompatible with the comprehensive sequencing and homology data presented in Lau’s previous seminal 2013 study. Significantly, the spurious claim, attributed specifically to Littlechild and Isupov, was subsequently withdrawn [88].

## 7. As the Area of Light Expands, So Does the Perimeter of Darkness

Further insights into the newly recognised status of 2,5-DKCMO as an fd-TCMO then emerged from subsequent studies that scrutinised the nature and extent of those FR activities able to support the ketolactonase throughout the successive trophophasic and idiophasic stages of growth of *P. putida* ATCC 17453 in a camphor-based minimal medium [90,91]. While supporting the finding of the 2013 report [87] that Fred was an inducible chromosome-coded dimeric (2 × 18 kDa) nonflavoprotein reductase, these new investigations revealed that it was only present in detectible levels in late trophophase (primary metabolism) and subsequent idiophase (secondary metabolism). In this respect, it resembles the equivalent ‘tailoring FRs’ involved in the late stage synthesis of some antibiotics [92,93], and a wide range of other microbial secondary metabolites [94,95,96,97,98,99]. Circumstantial evidence also suggests that Fred may correspond to the NADH oxidase activity of camphor-grown *P. putida* ATCC 17453 previously reported by both Gunsalus [28,29,30,31,32,33,34,35,36,37,38] and Trudgill [47,48,49,50,51,52,53], who both focussed their research exclusively camphor-grown cells entering idiophasic growth, a time frame chosen specifically to optimise biomass yield [52]. Trudgill conclusively confirmed by two different methods that the MW of the active form of the enzyme was 36 kDa, but significantly never subjected it to SDS-PAGE analysis.

Conversely, the equivalent role throughout trophophase (primary metabolism) was fulfilled by a combination of three newly recognised competent reductases. Two of these, Frp1 (27.0 kDa) and Frp2 (28.5 kDa), were constitutive monomeric nonflavoprotein ferric (flavin) reductases coded for by chromosomal DNA. Structurally, both corresponded closely with other well-characterised ferric (flavin) reductases and the related ferrodoxin reductases, a group of functionally related FRs that have been confirmed to be widely distributed in other prokaryotes [100,101]. Interestingly, the whole genome analysis of another strain of *P. putida* (KT2440, [102]) identified two such similarly sized activities (MW 35.0 kDa), subsequently named FprA and FprB [103,104,105]. Extensive kinetic studies of highly purified preparations of Frp1 and Frp2 from camphor-grown ATCC 17453 confirmed that they were both able to use NADH to reduce FMN very effectively, in each case by a sequential reaction mechanism [91].

By way of contrast, the third relevant activity after extensive purification was confirmed by sequence homology to be putidaredoxin reductase (PdR, MW 45.6 kDa), an inducible flavoprotein coded for by the *camA* gene on the CAM plasmid, and characterised by a tightly bound FAD cofactor [106,107,108,109]. This particular reductase was already known to function as one of the three activities (*camABC*) that collectively serve as cytochrome P450 monooxygenase (cytP450MO), the well-studied multienzyme complex that hydroxylates camphor [110,111,112], thereby initiating the terpene biodegradation pathway in *P. putida* ATCC 17453 (Figure 2). As the name implies, in that previously established context, PdR reduces putidaredoxin (Pdx, *cam*A), confirmed to occur by two successive one-electron transfers [112]. Given the novelty of this additional functionality of PdR in servicing 2,5-DKCMO, extensive relevant kinetic studies were deployed with highly purified preparations of both PdR and 2,5-DKCMO which conclusively confirmed this newly recognised NADH → FADH_2_ → FMNH_2_ role for PdR [91]. Considered in that specific context, these kinetic data implied that the bound FAD cofactor of the PdR could first receive and then subsequently donate two electrons, an outcome which correlates with the previously reported ability of purified PdR to function as a fully reversible NAD(H)-dependent dithiol/disulfide oxidoreductase with 5,5’-dithio-bis-(2-nitrobenzoic acid) [113]. In a broader context, the confirmed functionality between PdR and 2,5-DKCMO together with the previously established PdR-dependent mode of action of cytP450MO [110,111,112] implied that the bound FAD cofactor of the flavoprotein could promote two concurrent contrasting roles in the same bacterium, camphor-grown *P. putida* ATCC 17453 (Figure 7). Such duality of function is not unprecedented, and chimes with the growing recognition of the role of catalytic promiscuity as a feature of enzyme design and function [114,115,116,117,118,119].

The proven ability of PdR to satisfy the FMNH_2_ requirement of 2,5-DKCMO has additional strategic significance because it supports the concept that this key bio-oxygenating enzyme, responsible for initiating the ring cleavage of the bicyclic terpene (+)-camphor, can function independently of the chromosomal DNA of *P. putida* ATCC 17453. This contrasts with the previous proposals made by both Gunsalus and Trudgill that the monooxygenase is dependent on a 36 kDa chromosome-coded NADH oxidase as its source of FMNH_2_ (vide supra). This aspect of the functional independence of 2,5-DKCMO from chromosomal DNA was then supported further by using the known inhibitory effect of Zn^2+^ on ferric (flavin) reductases [120,121,122,123]. Again, relevant designed and executed kinetic studies were used to demonstrate that the lactone-forming activity of the monooxygenase was unaffected by the absence of functioning Frp1 and Frp2 activities [124].

Given this now-appreciated fuller understanding of the multiplicity of competent FMNH_2_-generating enzymes that can service the bio-oxygenating activity of 2,5-DKCMO in (+)-camphor-grown *P. putida* ATCC 17453, a valid reflexion to consider is why did neither Gunsalus or Trudgill identify the now-recognised redox significance of either Frp1 and/or Frp2, and/or PdR for the ketolactonase? It seems likely that the reported exclusive reliance [52] by both these researchers on isolating relevant enzymes from cultures of the bacterium entering the idiophasic growth in camphor-based minimal medium is probably the most likely explanation for their shared significant oversight in this respect.

Currently there still remain a number of incompletely understood or unresolved aspects of the bio-oxygenating activity of 2,5-diketocamphane 1,2-monooxygenase from camphor-grown *P. putida* ATCC 17453. One important issue is to develop a more comprehensive understanding of the biochemical logistics that enable PdR to serve both as a redox partner for 2,5-DKCMO and as an integral member of the trimeric cytP450MO bio-oxygenating consortium (Figure 7). Also requiring further investigation is the possibility that there may be other as yet incompletely characterised FMNH_2_-generating chromosome-coded FRs present in camphor-grown ATCC 17453 corresponding either to FprA and FprB of *P. putida* KT2440 [102,103,104,105], or the widely distributed but poorly characterised dodecameric ‘sulfite reductase’ [125,126,127,128]. In this respect, the SDS-PAGE analysis of highly purified extracts prepared from camphor-grown *P. putida* ATCC 17453 harvested throughout trophophase confirmed the presence of some proteins other than PdR, Frp1, and Frp2 which exhibited very low but detectible FR titres under the limited range of conditions tested to date. It is possible that two of these detected proteins (MW 33.0 kDa and MW 34.5 kDa) may indeed correspond to the similarly sized FprA and FprB (MW 35.0 kDa) FR activities of *P. putida* KT2440. However, the nature and function of these various incompletely characterised FR-positive activities awaits further investigation and clarification.

A further interesting avenue worthy of additional future investigation is the recently confirmed of ability of non-native FRs from other bacterial species to serve as proficient sources of the FMNH_2_ cosubstrate to promote nucleophilic bio-oxygenation by 2,5-DKCMO [129]. Confirmed competent FRs included not only the previously recognised NADH-dependent Fre from *E. coli* [83,84], but also other non-native FRs including FRD_Aa_. This dimeric (2 × 44.0 kDa) NADH-dependent FR from *Aminobacter aminovorans* ATCC 29600, which had previously been reported as supporting the fd-TCMO nitrilotriacetate monooxygenase [130], proved to be >30% more efficient than either Frp1 or Frp2 as a partner for promoting bio-oxygenation by 2,5-DKCMO. This aspect of inter-species redox coupling has significant potential practical implications for optimising the recognised applicability of 2,5-DKCMO as a biocatalyst in undertaking the bio-oxygenations of practical and commercial value [16].

So, more than six decades after Gunsalus’ pioneering studies reported 2,5-diketocamphane 1,2-monooxygenase as the first ever confirmed Baeyer–Villiger monooxygenase, thereby establishing the vital role that dioxygen plays in alicyclic ring fission by camphor-grown *P. putida* ATCC 17453, the biochemistry that accomplishes that key bio-oxygenation step remains incompletely resolved, and continues to require fuller understanding and delineation. Such is the transitional truth of knowledge.

## Figures and Tables

**Figure 1 microorganisms-12-00389-f001:**
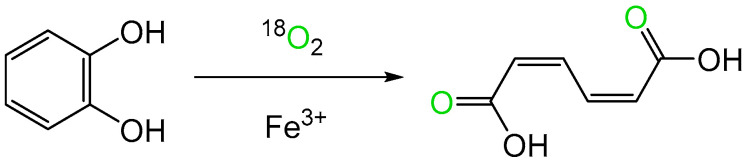
Bio-oxygenation of catechol to cis,cis-muconate by ‘pyrocatechase’ (catechol 1,2-dioxygenase) from a cell-free extract of a microbial soil isolate.

**Figure 2 microorganisms-12-00389-f002:**
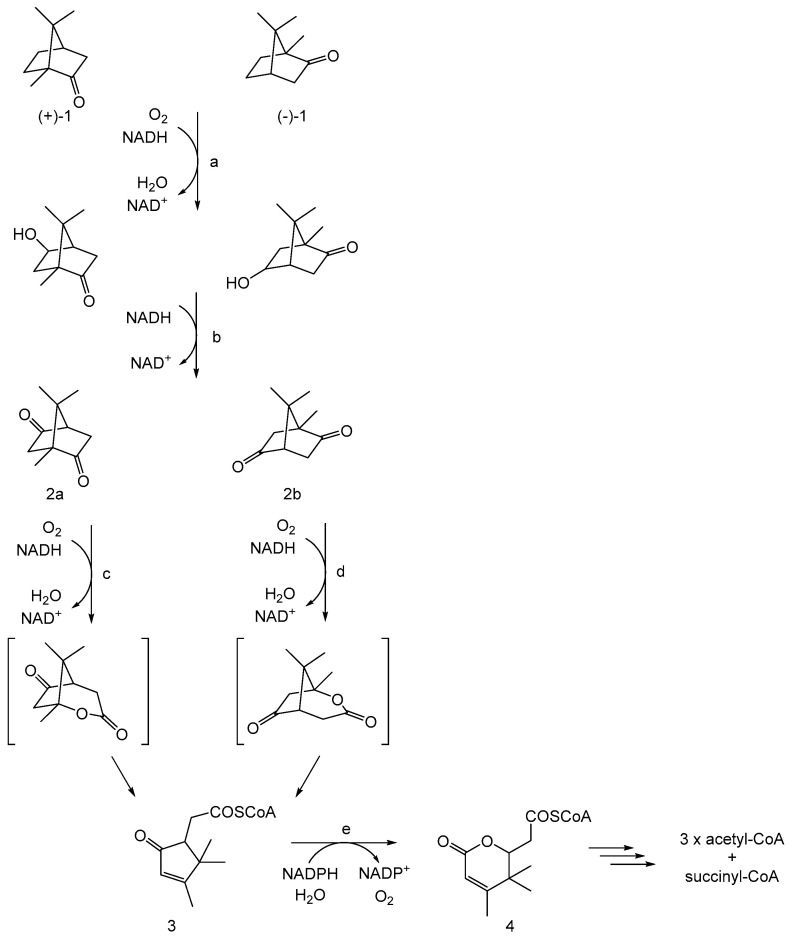
CAM plasmid-coded pathway of (*rac*)-camphor degradation by *Pseudomonas putida* ATCC 17453: adapted from [16]. a = cytochrome P450 monooxygenase (*camCAB*); b = hydroxycamphor dehydrogenase (*camD*); c = 2,5-diketocamphane 1,2-monooxygenase (*camE*_25-1_ + *camE*_25-2_); d = 3,6-diketocamphane 1,6-monooxygenase (*camE*_36_); e = 2-oxo-Δ^3^-4,5,5,-trimethylcyclopentenylacetyl-CoA monooxygenase (*camG*); (+)-1 and (-)-1 = (*R*)- and (*S*)-enantiomers of camphor, respectively; 2a = 2,5-diketocamphane; 2b = 3,6-diketocamphane; 3 = 2-oxo-Δ^3^-4,5,5-trimethylcyclopentenylacetyl-CoA; 4 = 3,4,4-trimethyl-Δ^2^-pimelyl-CoA lactone.

**Figure 3 microorganisms-12-00389-f003:**
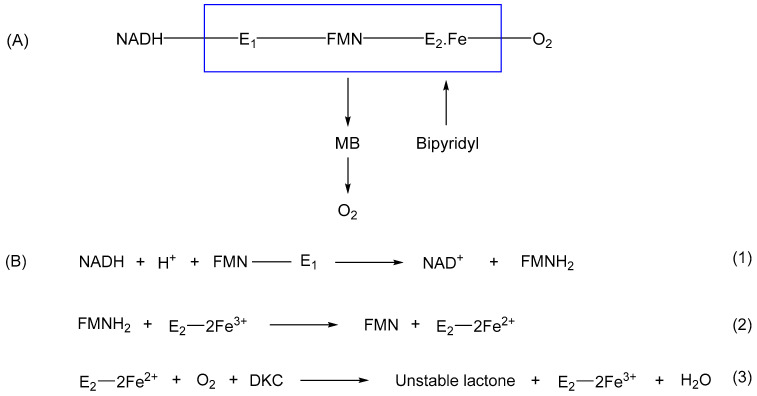
(**A**,**B**) The two representations developed by Gunsalus [12,13,14,15,16,17] to explain the functional interaction of NADH oxidase and 2,5-diketocamphane monooxygenase in undertaking O_2_-dependent lactonization of 2,5-diketocamphane in camphor-grown *Pseudomonas putida* ATCC 17453. E_1_ = NADH oxidase; E_2_ = 2,5-diketocamphane monooxygenase; MB = methylene blue; DKC = 2,5-diketocamphane.

**Figure 4 microorganisms-12-00389-f004:**
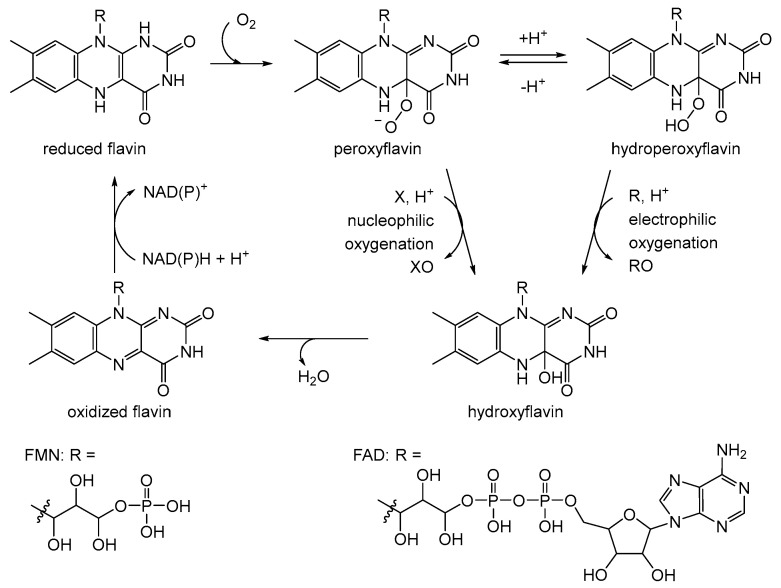
Generalised summary of the reactions undergone by flavin nucleotide cofactors of canonical flavoprotein monooxygenases in incorporating molecular oxygen as a participating reactant. X = ketone; XO = lactone/aldehyde; R = organosulfide; RO = sulfoxide.

**Figure 5 microorganisms-12-00389-f005:**
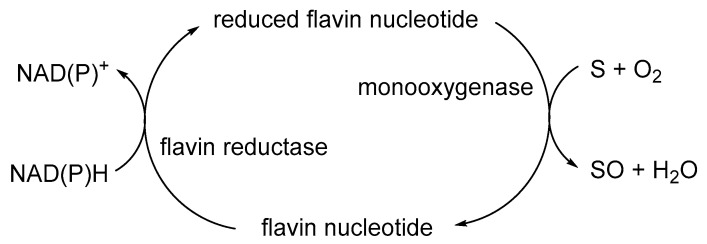
Generalised summary of the two clearly distinct reductive and oxidative half-reactions that characterise flavin-dependent two-component monooxygenases in incorporating molecular oxygen as a participating reactant.

**Figure 6 microorganisms-12-00389-f006:**
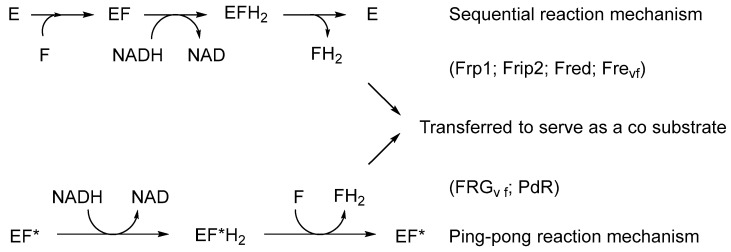
The different types (bound vs. unbound flavin) and different reaction mechanisms (sequential vs. ping-pong) of flavin reductases. E = nonflavoprotein flavin reductase; EF* = flavoprotein flavin reductase; F = FMN.

**Figure 7 microorganisms-12-00389-f007:**
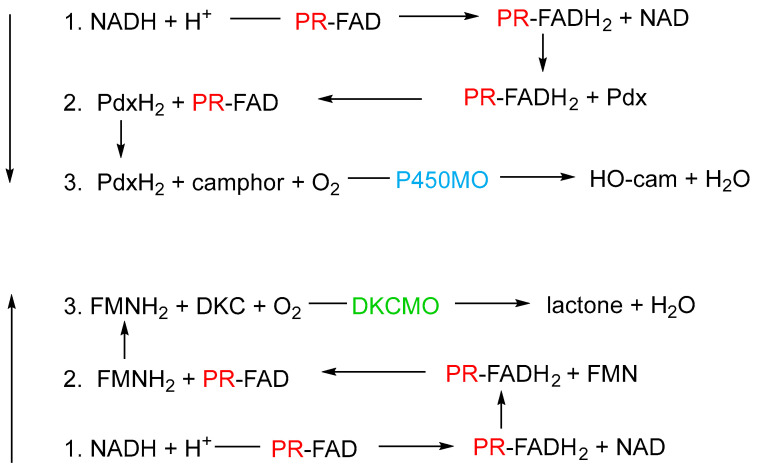
Schematic of the two contrasting roles of putidaredoxin reductase (PR) in camphor-grown *P. putida* ATCC 17453. Reaction 1 is common to both roles. In both cases, the sequence of reactions progresses from one to three. P450MO = cytochrome P450 mono-oxygenase; Pdx = putidaredoxin; DKCMO = 2,5-diketocamphane 1,2-mono-oxygenase; DKC = 2,5-diketocamphane.

## Data Availability

Not applicable.

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
