# Peer review of "The Role of Dioxygen in Microbial Bio-Oxygenation: Challenging Biochemistry, Illustrated by a Short History of a Long Misunderstood Enzyme"

_microorganisms, 2024, doi:10.3390/microorganisms12020389_

Round 1

Reviewer 1 Report

Comments and Suggestions for Authors

The manuscript by Andrew Willetts provides a relatively brief review of the literature on the role of dioxygen in enzyme-mediated microbial biocatalysis and biodegradation. Despite its small size, the manuscript contains references to 125 literary sources. The evolution of scientists' views on the role of dioxygen in microbial biocatalysis and biodegradation is examined in sufficient detail. From this point of view, it seems to me that the manuscript is very useful for researchers in this field and can be published in “microorganisms”.

I am not a microbiologist and have worked on the processes of hydrocarbon biodegradation as a geochemist, but always in close collaboration with microbiologists. Therefore, my assessment of this work is that of a geochemist, not a microbiologist.

The author absolutely correctly noted in the abstract that dioxygen-dependent microbial enzymes play in facilitating key reactions that drive the global geochemical Carbon Cycle. However, further in the manuscript the global carbon cycle is not considered in any way, although this would be an important applied conclusion from this review work. On the other hand, I understand that the manuscript is largely devoted to specific microbial processes of biocatalysis and biodegradation, and does not have the task of considering biodegradation processes more broadly in their applied significance. However, at the conclusion of the manuscript, it might make sense to add at least a couple of sentences about the applied significance of dioxygen in microbial biocatalysis and biodegradation for the global carbon cycle, etc.

A few minor notes on the text:

line 57: remove comma in reference [9].

Figure 1: Move the arrow to the right of catechol.

Lines 127-129: Remove blue lines from text.

Line 145: The sentence is broken like a paragraph, although it continues at line 153.

Line 320. Place a point instead of a comma after references to [104-106].

Author Response

Reviewer 1.

  1. The title of the review has been amended to more correctly reflect emphasise the biochemistry of dioxygen involvement with a specific microbial monooxygenase (2,5-DKCMO) rather than any implied involvement with biodegradation in general. Additionally, the final section of the review has now been revised extensively, both to make it simpler to comprehend and also to address potential applications of this type of enzyme for biocatalysis. Collectively, these changes make a more fitting conclusion to the review that better matches the revised running title.
  2. Each of the few minor notes have been addressed and the text amended accordingly.

Reviewer 2 Report

Comments and Suggestions for Authors

Dear Author!

In my opinion, the presented manuscript is devoted not so much to elucidation of the role of molecular oxygen in microbial biocatalysis and biodegradation (such a global topic can hardly be revealed in a small review), but rather to philosophical aspects of any scientific research on the example of evolution of knowledge about one enzyme - 2,5-diketocamphane-1,2-monooxygenase (EC 1.14.14.108). Discoveries, mistakes, failures, and discoveries again - all this is the thorny path of scientists or, as the author writes, "a paradigm of the ever-evolving nature of scientific knowledge". You have vividly illustrated this path. In this respect, the mini-review could be useful to every developing scientist, especially young scientists.

The manuscript is written in artistic language and reads with great interest as a detective novel, while the material is presented at a high scientific level, both from the standpoint of chemistry and related fields. My first reaction was that this manuscript should be published. I still think so, but I have doubts. I would like to express some doubts.

You published an excellent review of "The Isoenzymic Diketocamphane Monooxygenases of Pseudomonas putida ATCC 17453-An Episodic History and Still Mysterious after 60 Years" in Microorganisms in 2021! So, the subject in both reviews is the evolution of knowledge about the biochemistry of the same enzyme. Of course, much of the references overlap.

I also had some comments while reading the manuscript.

1. Figure 1 needs to be redone. I recommend presenting the reaction scheme and highlighting the embedded oxygen atoms into the cis,cis-muconate.

2. Figure 2 is in poor quality. In addition, the biochemical pathway becomes uninformative if the metabolites are listed as names rather than formulas.

3. Table 2 is formal and uninformative.

4. Almost all diagrams (Figures 3-7) are poorly done.

Author Response

Reviewer 2.

  1. I agree absolutely with the initial summary of Reviewer 2. As also emphasised by Reviewer 3, the intention of the review is to provide a more philosophical account of the progression of ideas that has led to our current understanding of the mode of 2,5-DKCMO. In doing so, it introduces both pre-2021 and post-2021 material of considerable significance not included in the previous 2021 Microorganisms review to specifically emphasise not only things that researchers got right, but also conversely things they got wrong. As far as possible from the extant record, it places into context the significance of both these types of defining activities (e.g. role of Mossbauer spectroscopy [Helmut Beinert], and the correlatio between ketolactonases and bacterial luciferases [Raffaella Villa]).
  2. All the Figures have been reformatted to a higher and more consistent quality. Table 1 (which was originally drawn up as an aide memoire to myself) has been replaced with  corresponding text entries in context.

Reviewer 3 Report

Comments and Suggestions for Authors

The manuscript entitled “The role of dioxygen in microbial biocatalysis and biodegradation - challenging biochemistry, illustrated by a short history of a long misunderstood enzyme aims reviewing the evolution of the current status of 2,5-diketocamphane 1,2-15 monooxygenase (EC 1.14.14.108) from camphor-grown Pseudomonas putida ATCC 17453. To my mind this manuscript is topical and corresponding to the aims and scopes of the “Microorganisms” journal.

This review contains important insight into the role of oxygen in the metabolism of microorganisms and the evolution of understanding of the process of using molecular oxygen in zyme-catalysed reaction since the mid-1930s. I would especially like to note the important philosophical aspects of this process noted by the author, which is quite rare in modern review articles. This review is a fundamental, well-thought-out text and I have only few comments on its merits. I had great pleasure while getting acquainted with this manuscript. It is really good both from the side of organic chemistry and from the side of biochemistry.

The notes for the text are as follows:

1.     The quality of the diagram shown in Figure 2 should be improved.

2.     Please remove the red underlining in Figure 4 and decipher the letter designations in the captions to the figures.

3. Please remove the red underline in Figure 7 and decipher the color designations in the captions to the figures

4. Please separate the conclusion into a subparagraph and try to make it simpler and more philosophical. This will make your work more understandable for classical microbiologists who are not deep specialists in the field of biochemistry.

Author Response

Reviewer 3.

  1. I would like to thank Reviewer 3 for the insightful and kind summary comments, which clearly resonate with Reviewer 2. At my age I think I'm allowed to be a little philosophical about things!
  2. The concluding section of the text has now been revised and reformatted to make it simpler to understand and to be a more fitting conclusion to the review.
  3.  Figures 2 and 4 have been amended as requested.

Reviewer 4 Report

Comments and Suggestions for Authors

Manuscript ID microorganisms-2805167 entitled "The role of dioxygen in microbial biocatalysis and biodegradation–challenging biochemistry, illustrated by a short history of a long misunderstood enzyme."

In this Review, the authors reported the important roles that mono- and di-oxygenases play in relevant aspects of bio-oxygenation, the global geochemical carbon cycle, and microbial biocatalysis and biodegradation. The author shows interesting reports and results. However, revisions would be needed to have the manuscript may be considered for publication in Microorganisms.

My comments:

1. It is suggested that the author remove the “–” from the title and change it to “:” “The role of dioxygen in microbial biocatalysis and biodegradation: Challenging biochemistry…”

2. Page 2, line 48. What does the author mean by O2-18? He speaks of the isotope of oxygen since the natural element would be O2-16 (mass 16).

3. Page 2, line 72. Put the Figure after what is mentioned (Fig. 2).

4. Correct Figure 1.

5. Page 3. Improve the quality of Figure 2, chemical structures and letters.

6. Is Figure 2 a mechanism proposed by the author, or is it already published? It is recommended to indicate if it has already been published in another journal (add Figure and reference permission), the same for the other figures presented.

7. The text lacks a discussion on Fig. 2; it is recommended to add a discussion since it is only mentioned.

8. Page 4, Table 1. It is recommended to reorganize Table 1, add other columns for author or year and another column for references (check the journal Tables format).

9. Page 4, lines 141-142. It is recommended that the paragraph be rewritten and the word "cartoon" changed.

10. Page 5, line 169. In some cases, the author uses, eg, in others, e.g., homogenize.

11. Page 5, lines 175-178. It is recommended to rewrite the paragraph. It needs to be clarified.

Is the written word “de-ja vu” correct?

“Despite these various advances, most of Trudgill's limited output over the period 1982 -93 [47 -53] had more than a hint of de-ja vu, being essentially a reexamination of structural aspects of 2,5-DKCMO (E2) and its associated NADH oxidase activity (E1).”

12. Page 6, line 193. The author is suggested to rewrite title 5, “5. New insights–the various roles of serendipity,” which is unclear. And use “:” instead of “-“.

13. Page 7, Figure 6. Review the Figure and correct. “cosubstrate."

14. Page 9, line 304. What does the author mean by title 7? “7. As the pool of light expands, so does the surrounding halo of darkness." editing is recommended.

15. Page 9, correct Figure 7, text.

16. The title is "The Role of Dioxygen in Microbial Biocatalysis and Biodegradation." The biodegradation part, or where it has currently been applied in organic compounds and plastics, needs to be discussed.

A paragraph or text on biodegradation should be added, where mono-and di-oxygenase have participated. An important application today due to pollution is in the biodegradation of plastics.

Part of the text and the following current references are recommended to discuss and support this part:

The participation of the enzyme through oxidative activities (mono-, dioxygenase) in the fragmentation and assimilation of different plastics has been published to contribute to the biodegradation of plastics [1,2].

[1] Heliyon 2023, 9:e21374.  https://doi.org/10.1016/j.heliyon.2023.e21374

[2] Chemosphere 2022, 307:136136. https://doi.org/10.1016/j.chemosphere.2022.136136

and other references.

17. Add conclusions and/or perspectives.

What are the main applications?

What has been done or reported on biodegradation?

What are the limitations that currently exist?

What is missing in the topic to be developed?

What is suggested for future research?

18. Current references (2021-2023) are recommended.

References need to be updated since a review was presented and must be updated by 2023. No references from the last 3 years (2021-2023) are presented; only one from 2022 and one from 2023 are presented.

Comments on the Quality of English Language

Moderate editing of the English language is required.

Author Response

Reviewer 4.

  1. The title of the review has been changed to more correctly reflect the narrow focussed attention of the text on the biochemistry that defines the relationship between dioxygen and a specific microbial monooxygenase. This now signals that the review is not intended to address those much broader considerations relevant to aerobic biodegradation (comments relevant to point 16 as well).
  2. 18Ois the isotope of dioxygen with a MW of 18. Its value is in tracing where the oxygen atoms are introduced in generating cis,cis-muconate.
  3. Amended. 
  4. Amended.
  5.  Amended.
  6.  Figure 2 is an amended form of the equivalent data originally included in Willetts, A., Microorganisms 20197, 1.
  7.  See 6.
  8. Table 1 has been replaced with text equivalent.
  9.  Amended.
  10.  Amended.
  11. deja vu (literally = 'aleady seen' [French]) is accepted contemporary usage in this context.
  12.  Amended.
  13. Cosubstrate is the correct terminology to use in this context - see Ellis, H. [64].
  14. The title of the section has now been amended to a direct aphorism widely attributed to Albert Einstein - it represents a philosophical concept that acknowledges the limitations of established knowledge.
  15.  Amended.
  16. See response 1 above.
  17.  The concluding section of the review (last few paragraphs) has been amended and extended along the lines suggested.
  18.  Additional references for 2020 - 2023 have now been introduced in text.

Round 2

Reviewer 4 Report

Comments and Suggestions for Authors

The author made the suggested changes according to the reviewers' comments for Manuscript ID microorganisms-2805167. The paper is within the scope of the journal. Thus, the manuscript is suitable for publication in Microorganisms.